# Picks in the Fabric of a Polyploidy Complex: Integrative Species Delimitation in the Tetraploid *Leucanthemum* Mill. (Compositae, Anthemideae) Representatives

**DOI:** 10.3390/biology12020288

**Published:** 2023-02-10

**Authors:** Christoph Oberprieler, Tankred Ott, Robert Vogt

**Affiliations:** 1Evolutionary and Systematic Botany Group, Institute of Plant Biology, University of Regensburg, Universitätsstr. 31, D-93053 Regensburg, Germany; 2Botanic Garden & Botanical Museum Berlin, Freie Universität Berlin, Königin-Luise-Str. 6-8, D-14191 Berlin, Germany

**Keywords:** Asteraceae, ecology, ddRADseq, geography, leaf morphology, nomenclature, polyploidy, taxonomy

## Abstract

**Simple Summary:**

The delimitation of species as the most important rank in biological classification is an essential contribution of taxonomy to biodiversity research, with all of its evolutionary, ecological, political, and legislative ramifications. Species delimitation is extremely tricky in plant groups evolving by polyploidisation (multiplication of chromosome sets) because the rapid formation of new, reproductively isolated lineages (species) is often not paralleled by conspicuous genetic, morphological, physiological, and/or ecological differentiation. Having clarified the taxonomy of diploid (2*x*) representatives of the genus *Leucanthemum* (marguerites, ox-eye daisies) in a previous contribution, the present study aims at an objective and reproducible delimitation of evolutionarily significant units (species) at the tetraploid (4*x*) level. We used DNA-based fingerprinting and statistical analyses of leaf shapes, ecological niches, and distribution ranges for eight predefined morphotaxa to judge their ranks as species or subspecies and propose a taxonomical treatment for the surveyed group with six species (two of them with two subspecies). Having clarified the taxonomic structure of the ancestral diploid (the ‘warps and wefts’) and the subsequent tetraploid layer (the ‘picks of the fabric’), we will be able to provide a taxonomy for the remainder of this well-known plant group and study its reticulate evolutionary history.

**Abstract:**

Based on the results of a preceding species-delimitation analysis for the diploid representatives of the genus *Leucanthemum* (Compositae, Anthemideae), the present study aims at the elaboration of a specific and subspecific taxonomic treatment of the tetraploid members of the genus. Following an integrative taxonomic approach, species-level decisions on eight predefined morphotaxon hypotheses were based on genetic/genealogical, morphological, ecological, and geographical differentiation patterns. ddRADseq fingerprinting and SNP-based clustering revealed genetic integrity for six of the eight morphotaxa, with no clear differentiation patterns observed between the widespread *L. ircutianum* subsp. *ircutianum* and the N Spanish (Cordillera Cantábrica) *L. cantabricum* and the S French *L. delarbrei* subsp. *delabrei* (northern Massif Central) and *L. meridionale* (western Massif Central). The inclusion of differentiation patterns in morphological (leaf dissection and shape), ecological (climatological and edaphic niches), and geographical respects (pair-wise tests of sympatry vs. allopatry) together with the application of a procedural protocol for species-rank decisions (the ‘Wettstein tesseract’) led to the proposal of an acknowledgement of the eight predefined morphotaxon hypotheses as six species (two of them with two subspecies). Nomenclatural consequences following from these results are drawn and lead to the following new combinations: *Leucanthemum delarbrei* subsp. *meridionale* (Legrand) Oberpr., T.Ott & Vogt, comb. nov. and *Leucanthemum ruscinonense* (Jeanb. & Timb.-Lagr.) Oberpr., T.Ott & Vogt, comb. et stat. nov.

## 1. Introduction

The delimitation of species as the most paramount rank in biological classification is an essential contribution of taxonomy to biodiversity research, with all of its evolutionary, ecological, political, and legislative ramifications and corollaries. Following Stuessy [1] and Zachos et al. [2], this alpha-taxonomy procedure has a twofold nature: while in the first step (the ‘grouping’ step), ‘species discovery’ and ‘species validation’ methods are used to infer and subsequently test species-group hypotheses (the ‘species taxa’ of Zachos et al. [2]) by detecting genealogical, morphological, or ecological discontinuities [3,4], the second step (the ‘ranking’ step) constitutes ‘an executive decision that the species taxon warrants recognition at the species level’ [2]. However, the latter—the decision whether ‘species taxa’ should be ranked as species under the Linnaean classification system—is clearly subjective, due to its dependence on the acceptance of a species concept. Of these, a broad array exists, though without any cognisable chance for the unrestricted applicability of a single one throughout the realm of organismic diversity.

The ‘unified species concept’ proposed by De Queiroz [5] was a game-changer in the futile search for a generally applicable species concept because it altered the perspective that the properties entertained by the plethora of concepts (e.g., reproductive isolation, genealogy, morphology, ecology, geography, etc.) are helpful in species conceptualisation. Instead, the ‘unified species concept’ defined species as hypothetical independently evolving metapopulation lineages, for which the above-mentioned properties could be made to subserve as indicators or proxies. This conceptual shift paved the way for the renaissance of ‘biosystematics’ or ‘experimental taxonomy’ approaches to species delimitation used in the second half of the 20th century as ‘integrative taxonomy’ [6,7], which makes use of all available sources of empirical evidence for the conceptualisation of species rank. This is nowadays carried out either by using computational tools, e.g., Geneland [8] for the joint analysis of morphology, genetics, and geography; “multivariate normal mixtures and tolerance regions” analysis [9,10] for morphology and geography; iBPP [11] for genealogy and morphology; regression analysis [12] for genetics and geography; or by entertaining procedural protocols, e.g., [13,14,15,16]. Most recent approaches also try to incorporate the speciation process itself into species-delimitation software programs (Delineate [17]).

Species formation by polyploidy—a speciation mode realised in significant numbers of pteridophyte and angiosperm groups—poses considerable problems for species delimitation [18]: while polyploidisation will instantly lead to postzygotic reproductive isolation between parental taxa and their polyploid derivatives (biological species concept), detectable trait differences entertained by a morphological or a physiological/ecological species concept may only be realised in an allopolyploid speciation scenario and not (or at least not immediately or obviously) in an autopolyploid one. Additionally, a strict phylogenetic species concept (with species defined as monophyletic evolutionary units) is violated both in the auto- and allopolyploid speciation mode due to (a) the parental species becoming paraphyletic relative to the newly formed polyploid species and (b) the potentially polyphyletic nature of a polyploid species caused by its multiple (and sometimes reciprocally parented) origin or (c) gene flow between independently formed allo- and/or autopolyploid populations/lineages. Finally, at least in some plant groups, the switch of polyploid lineages toward asexual reproduction (agamospermy) makes the application of a biological species concept senseless. As a consequence, these peculiarities of polyploid species formation make the application of a ‘unified species concept’ sensu De Queiroz [5] (species as metapopulation lineages) and the integrative approach to species delimitation indispensable for plant groups diversifying through this speciation mode.

The genus *Leucanthemum* Mill. (Compositae, Anthemideae; marguerites or ox-eye daisies) comprises 39 [19] to 42 species [20], with ploidy levels ranging from diploid (2*x*) to dodecaploid (12*x*), and one species [*L. lacustre* (Brot.) Samp.) from Portugal even showing a chromosome number of 2*n* = 22*x* = 198 (docosaploid level). *Leucanthemum* is distributed over the whole European continent and extends into Northern Asia (the tetraploid *L. ircutianum* DC. is found in Siberia), while some species were also introduced into temperate regions of the Northern and Southern Hemispheres [21]. The current species delimitation of the genus is mostly based on differences in morphology, especially in general leaf shape and leaf dissection, as the flower characters are relatively invariant, and ploidy level, as well as geographical distribution [22,23] and, more recently, genetic differentiation [19].

Previous studies addressed the species delimitation and phylogeny of the diploid *Leucanthemum* representatives based on multicopy nuclear markers (nrDNA ETSs), AFLP fingerprinting, and single-copy nuclear markers [24,25,26,27]. Cloning of nrDNA ETS amplicons revealed that some diploid taxa exclusively possess a plesiomorphic ETS ribotype cluster closely related to ETS ribotypes of the outgroup and that others are characterised by the exclusive possession of an apomorphic ETS ribotype cluster, while a third group of taxa exhibit an additive pattern of the two types [24]. This finding was supported by Konowalik et al. [25] using AFLP fingerprinting and multilocus species-tree reconstructions based on low-copy markers of the nuclear genome, and it was demonstrated that the species with the plesiomorphic ETS ribotypes form an early-diverging paraphyletic grade, in which the monophyletic group of taxa with the apomorphic ETS ribotypes are nested. Furthermore, the authors applied coalescent-based simulations to distinguish between hybridisation and incomplete lineage sorting (ILS), revealing that for most of the diploid taxa involved (and especially for those in the second group) incongruence among gene trees could not be explained by ILS alone and that recent hybridisation or even homoploid hybrid speciation events must be assumed. Consequently, some infraspecific taxa were raised to species level to account for their assumed independent formation through homoploid hybrid speciation (i.e., *L. cacuminis*, *L. eliasii*, and *L. pyrenaicum*).

Species delimitation in a morphologically close-knit group of taxa in the clade with the apomorphic nrDNA ETS ribotypes was subsequently carried out by Wagner et al. [26], who used AFLP fingerprinting, sequence information from plastid and nuclear low-copy markers, and coalescent-based Bayesian delimitation methods to infer species boundaries in the *L. ageratifolium* group, despite the frequent presence of hybrid individuals in this group. Finally, Wagner et al. [27] presented a multilocus phylogenetic reconstruction of the subtribe Leucantheminae, in which the diversification among diploid *Leucanthemum* species was dated to the last 1.9 (1.1–2.9) Ma, arguing for the strong influence of Pleistocene oscillations on species formation in this genus. Most recently, Ott et al. [19] combined restriction-site-associated DNA sequencing (RADseq), ecological niche modelling (ENM), geographical patterns, and geometric morphometrics for integrative species delimitation among the diploid *Leucanthemum* representatives.

In contrast to the extensive investigation of species delimitation and the phylogenetic relationships of diploids, the evolutionary histories of only a few tetraploid *Leucanthemum* species have been the subject of previous studies. Oberprieler et al. [28] found indications of an allopolyploid origin of the tetraploid *L. ircutianum* subsp. *ircutianum* based on AFLP markers. One year later, Greiner et al. [29] showed that the tetraploid *L. pseudosylvaticum* is able to form fertile offspring when crossed with its diploid relative *L. pluriflorum*, suggesting a relatively recent diversification. Another study of the *L. pluriflorum* group by Greiner et al. [30] produced evidence for the allopolyploid and autopolyploid origins of *L. pseudosylvaticum* and *L. corunnense*, respectively, and raised *L. pseudosylvaticum*, which was formerly a subspecies of *L. ircutianum,* to species rank. Finally, the most recent contribution to the taxonomy of the tetraploid *Leucanthemum* representatives by Oberprieler et al. [31] suggested infraspecific ranks for the widely distributed *L. ircutianum* subsp. *ircutianum* and its amphi-Adriatic counterpart *L. ircutianum* subsp. *leucolepis* based on AFLP fingerprinting and sequence variations in nuclear and plastid DNA.

The present contribution aims at a comprehensive and integrative assessment of all tetraploid *Leucanthemum* taxa. Applying the conceptual framework for species-rank decisions proposed by Oberprieler [16], morphological, ecological, geographical, and genealogical/genetic evidence is used to devise a taxonomic treatment for eight tetraploid taxa that have been hitherto accepted as occupying specific or infraspecific ranks in Central and Southern Europe, the only exception being the NW Spanish *L. corunnenese* Lago, for which an autotetraploid origin based on the diploid *L. pluriflorum* has been shown in a previous study [30].

## 2. Material and Methods

### 2.1. Taxon Selection

Besides the widespread *L. ircutianum* subsp. *ircutianum,* the following seven geographically more restricted tetraploid taxa were considered as entities, for which an integrative taxonomic treatment was envisioned in the present contribution: these comprise the three species endemic to the Iberian Peninsula, *L. cantabricum* Sennen, *L. crassifolium* (Lange) Lange, and *L. pseudosylvaticum* (Vogt) Vogt & Oberpr., along with the NE Spanish and SW French *L. delarbrei* Timb.-Lagr. subp. *delarbrei,* subsp. *ruscinonense* (Jeanb. & Timb.-Lagr.) Vogt et al. and *L. meridionale* Legrand and the amphi-Adriatic *L. ircutianum* subsp. *leucolepis* (Briq. & Cavill.) Vogt & Greuter (see Figure 1 for a distribution map).

### 2.2. Morphological Analyses

Differences in leaf shape, an important delimitation criterion for *Leucanthemum* taxa, have been demonstrated in the diploids of the genus [19] and were assessed here by measuring the general leaf shape and the degree of dissection of the leaves by applying elliptic Fourier analysis (EFA; [32]) and calculating leaf dissection indices (LDIs), respectively. For this purpose, 285 images of digitised herbarium specimens were provided by the herbarium of the Berlin Botanical Museum (B; see Appendix A). Using these images, we manually annotated 1,338 intact leaves with polygons around the leaves’ outlines and polylines along the leaves’ main veins using the Computer Vision Annotation Tool (CVAT; https://github.com/openvinotoolkit/cvat (accessed on 31 January 2023)) according to Ott et al. [19]. Subsequently, we straightened the leaves to reduce the influence of deformations either caused by developmental irregularities or introduced by the drying process and extracted the leaf contours as binary masks.

Using the binary masks, we conducted EFA and calculated LDIs using the Python packages scikit-image [33], numpy [34], scikit-learn [35], and PyEFD (https://github.com/hbldh/pyefd (accessed on 31 January 2023)). For the EFA, we used 20 harmonics, normalised the descriptors, and applied principal component analysis (PCA) for decorrelation and feature extraction to the non-constant descriptors (normalisation causes the three descriptors A1, B1, and C1 to be constant).

To find differences in general leaf shape, we used the first 15 principal components (PCs), explaining 50% of the total variance with two different testing strategies: (1) the permutation-based test for differences in Euclidean distance in PC space as applied by [19]; and (2) non-parametric multivariate analysis of variance (NPMANOVA; [36]) implemented in the R package ‘vegan’ v2.6 (function ‘adonis’; [37]). The number of permutations was set to 5000 and 100 for the former and the latter, respectively. To assess differences in leaf dissection, we subjected the LDI values to Welch’s tests. All tests were corrected using Bonferroni´s method.

### 2.3. Ecological Niche Modelling

Ecoclimatological and edaphic niches of the tetraploid *Leucanthemum* taxa were reconstructed using ecological niche modelling (ENM) and compared using permutation-based statistical tests. For this purpose, we retrieved the collection locations of 1470 individuals from several herbaria (Appendix A) and obtained rasters of 19 bioclimatic and 10 edaphic variables at depth levels of 0–5 cm, 5–15 cm, and 15–30 cm (Appendix A) from Worldclim Bioclim [38] and SoilGrids [39], respectively. The rasters were cropped to encompass Central Europe (longitude -10.0°–26.0°; latitude 35.9°–52°) and scaled to a resolution of 2.5 arc minutes using the R package ‘raster’ v3.5.15 [40] (https://github.com/rspatial/raster (accessed on 31 January 2023)). In addition to the climate rasters for presence, we retrieved paleoclimate rasters for the last glacial maximum (LGM; CCSM4) and the last interglacial (LIG; lig_30s) from Worldclim Bioclim and subjected them to the same preprocessing, but with additional recoding of temperature, since LGM and LIG datasets used Bioclim 1 temperature encoding. For the edaphic variables, we averaged the raster values of the three mentioned depth levels to obtain a single raster for each soil variable. Since it was computationally not tractable to work with all 29 rasters, we applied principal component analysis (PCA) with standardisation implemented in the R package ‘ENMtools’ v1.0.6 [41] for feature extraction to the recent Bioclim and SoilGrids rasters, separately. The first three principal-component (PC) rasters for each of the datasets were selected for the ENM.

To compare the ecological niches of the different taxa, we reconstructed potential distribution ranges for all tetraploid taxa except *L. meridionale* using MaxEnt v3.4.4 [42], with 5000 iterations and 6-fold cross-validation, and applied niche-equivalency tests implemented in ‘ENMTools’ for all combinations of tetraploid taxa with MaxEnt as species distribution, 200 replicates, a species range of 50 km, and 1000 background points with a range of 20 km. *Leucanthemum meridionale* was excluded from these comparisons because this species is endemic to a very limited area, causing all collection points to fall into the same raster cell and thus rendering ENM impossible. Potential niches at LGM and LIG were also reconstructed using MaxEnt with the same parameters as for the PC rasters, but this time with LGM and LIG as projection rasters and including collection data for the diploid species provided by Ott et al. [19].

### 2.4. Geographical Distribution

The geographic co-distribution of the tetraploid taxa, excluding *L. meridionale* again due to its restricted sampling area as a point endemic, was assessed by evaluating the overlap of spatial distributions using the same data as for the ENM analyses. Unfortunately, there are no comprehensive distribution rasters available for the tetraploid *Leucanthemum* representatives, which is why we had to resort to approximating the geographic distribution from the available sampling points. We applied the method proposed by Ott et al. [19], which approximates the true geographical distribution by reconstructing the potential area using an ENM (in this case, MaxEnt v3.4.4; [42]) and subsequently removes unconnected and unsampled regions (i.e., regions that are separated from collection points by areas of low probability). To test for sympatry, we applied the permutation approach of Ott et al. [19] with 400 simulated datasets and a threshold of 0.25; the resulting *p*-values were corrected for multiple testing using Bonferroni´s method.

### 2.5. RADseq Assembly

Double-digest RADseq (ddRADseq; [43]) was conducted based on an accession set comprising 51 individuals from all presently accepted 17 diploid *Leucanthemum* species (20 taxa; [19]) and 35 individuals from 8 tetraploid *Leucanthemum* taxa (see the table in Appendix A), the latter being the focal group of the present study. ddRADseq reads of the diploid samples were taken from Ott et al. [19], while for the tetraploid individuals, genomic DNA was extracted from silica-dried specimens according to the CTAB DNA extraction protocol of Doyle and Dickson [44]. For assembly-quality assessment, replicates were generated by repeating the extraction for two samples (*L. ircutianum* DC. subsp. *ircutianum*: accession L055-03/-031; *L. ircutianum* subsp. *leucolepis*: accession 170-02/-021). ddRAD Illumina sequencing (2 x 150 bp; NextSeq 500, Illumina Inc., San Diego, CA, USA) using the restriction enzymes *Pst1* and *ApeK1*, including demultiplexing and adapter clipping, was conducted by LGC Genomics (Berlin, Germany).

Demultiplexed and adapter clipped reads from the diploid and tetraploid samples were assembled using iPyrad v0.9.81 [45] against the reference provided by Ott et al. [19]; the minimum number of samples per locus (min_samples_locus) was set to 20, while the remaining parameters were kept at default values. For detection of assembly errors, locus, allele, and SNP error rates were calculated (see [19,46]).

### 2.6. RADseq Network Analysis

Neighbor-nets based on SNP-based Nei distances ([47]; https://github.com/simjoly/pofad (accessed on 31 January 2023)) were calculated using SplitsTree4 v4.15.1 [48] for (a) the complete dataset and (b) for the tetraploid accessions exclusively. Distances were calculated based on the variant output (VCF file) of iPyrad using a custom Python and C tool (https://github.com/TankredO/nei_vcf (accessed on 31 January 2023)). With SNP-based Nei distances, differences in variant (SNP) frequencies are directly included in distance calculation, rendering this kind of distance metric particularly useful for comparing diploid and polyploid samples.

### 2.7. RADseq Consensus Clustering

Weighted ensemble of random *(k)k*-means (WKM) clustering [49] was applied to detect clusters of genetically similar individuals within the group of tetraploid *Leucanthemum* representatives. WKM clustering is a semi-supervised consensus (ensemble) clustering technique, allowing for a priori-defined fuzzy pairwise must-link and must-not-link constraints. Roughly, WKM clustering fits a number of *k*-means clusterings (e.g., 1000), each for a random subset of features (‘variables’) and data points (‘samples’), and with a random number of clusters *(k)*. For each *k*-means run, clustering-level and cluster-level consistencies are calculated, which measure deviations from the fuzzy constraints concerning the whole clustering and single clusters, respectively. In addition, the mean silhouette index is calculated as an internal clustering-quality metric. The three values are non-linearly combined to obtain a clustering weight (i.e., a value determining the clustering quality). The clustering co-association matrix, i.e., the matrix determining which samples were clustered together (this can be thought of as a similarity matrix with only 1 and 0 entries; 1 if the samples were placed in the same cluster and 0 otherwise), is multiplied by the clustering weight. All weighted co-association matrices are summed (and optionally scaled) to obtain a consensus (co-association) matrix. Finally, a consensus clustering is calculated using this consensus matrix via hierarchical or spectral clustering.

For clustering of the tetraploid *Leucanthemum* representatives, the previously calculated SNP-based Nei distances (see above, under *Network analysis*) were subjected to a principal coordinate analysis (PCoA) with Lingoes negative eigenvalue correction, and the first principal coordinates (PCos) explaining at least 50% of the variance were selected. Based on the PCo scores, WKM clustering was applied with must-link constraints for the individuals of each of the taxa *L. cantabricum*, *L. crassifolium*, *L. delarbrei* subsp. *delarbrei*, *L. delarbrei* subsp. *ruscinonense*, *L. ircutianum* subsp. *leucolepis*, and L. *meridionale* and must-not-link constraints among individuals from *L. delarbrei* subsp. *delarbrei* and *L. meridionale*. The fraction of random features and samples was set to 0.60 and 0.80, respectively. The number of *k*-means runs was set to 5000, and *k* was allowed to take values from 2 to 14, including the lower and upper bounds. Finally, the consensus matrix was subjected to an average-linkage hierarchical clustering for each of the *k*-values, and the clustering scoring that was best according to the Bayesian information criterion (BIC) and the Calinski–Harabasz criterion (CH) was selected. All analyses were performed using pyckmeans v0.9.4 (https://github.com/TankredO/pyckmeans (accessed on 31 January 2023)).

### 2.8. Genealogical Species Delimitation

To find the potential diploid parental species of the tetraploid taxa under study, we applied a custom version of SNiPloid [50], as proposed by Wagner et al. [51]. Very similar to the original SNiPloid algorithm, our script compares two parent individuals (i.e., diploids) with one child individual (i.e., a tetraploid) on a per-SNP basis, distinguishing and counting the frequencies of five different SNP categories: categories 1 and 2 count so-called inter-specific SNPs, where exactly one parental SNP is identical to the child SNP and the other not. Categories 3 and 4 are so-called derived SNPs, meaning that an SNP is unique to the child individual, while the parents are homozygous for the same variant; patterns 3 and 4 cannot be distinguished based on unphased SNP data. Category 5 SNPs are homeo-SNPs, where the child is heterozygous (polymorphic), combining both parents’ homologous and monomorphic alleles. For all categories, only SNPs where both parents were homozygous (monomorphic) were considered.

We applied the SNiPloid approach to look for signs of allopolyploid speciation and expected recently formed allotetraploids to express a high number of category 5 SNPs, while longer established allotetraploid species should express a higher frequency of derived SNPs (i.e., category 3 and 4 SNPs). Finally, category 1 and 2 SNPs may indicate gene flow to the putative parents or alternatively be an additional signature of recent polyploidisation. For each tetraploid taxon and each possible pair of diploid parent species, therefore, we applied our custom Python implementation of the SNiPloid algorithm to the concatenated SNP output returned by iPyrad (.snps ouput). For each triplet, we tested all individuals and calculated the arithmetic means of SNP patterns 1 through 5.

## 3. Results

### 3.1. Morphology

The degree of leaf dissection varied significantly at an alpha level of *p*-value < 0.01 for all but three taxon pairs (Table 1, upper triangle). Differences in general leaf shape, measured as Euclidean distances among principal-component (PC)-transformed Fourier descriptors, were significant for 14 of the 28 taxon pairs (Table 1, lower triangle, left number). When applying NPMANOVA to the PC scores, 16 of the 28 taxon pairs showed significant differences in leaf-outline shape (Table 1, lower tringle, right number).

### 3.2. Ecological Niche Modelling and Geographical Range Overlap

All pairwise niche-equivalency tests except those for the two taxon pairs *L. cantabricum*–*L. crassifolium* (both N Spain) and *L. cantabricum* (N Spain)–*L. delarbrei* subsp. *ruscinonense* (NE Spain, SW France) were significant at an alpha level of 0.01 (see Table 2, where *p*-values of 0 indicate no niche overlap at all). Maps depicting predicted potential distribution ranges of the diploids and tetraploids based on recent, LGM, and LIG bioclimatic variables are provided in the supporting information (Appendix A).

In the tests on geographical range overlap (Table 3), all taxon pairs except for *L. cantabricum*–*L. crassifolium* (both N Spain) and *L. delarbrei* subsp. *ruscinonense* (NE Spain, SW France)–*L. ircutianum* subsp. *ircutianum* (large parts of Europe) were significant, suggesting a deviation from sympatric distributions for most of the tetraploid taxa.

### 3.3. RADseq Assembly and Analysis

Of 216,551,342 raw reads, 81,517,011 demultiplexed, adapter-clipped, and restriction-enzyme-filtered reads were mapped against the reference of Ott et al. [19]. The resulting assembly comprised 7,342 loci with 156,057 SNPs, of which 83,375 were parsimony-informative. The percentages of missing data for sequence and SNP matrices were 36.9% and 34.5%, respectively. The trustworthiness of the RADseq fingerprinting procedure was confirmed by the high degree of similarity between re-extracted and reanalysed accessions (L055-031, 170-021) and their counterparts (L055-03, 170-02) in the subsequent analyses.

The network reconstruction based on SNP-based Nei distances for the complete dataset (i.e., all diploid and tetraploid individuals) placed all tetraploid taxa into the so-called *L. vulgare* group (Figure 2A). While all diploid samples were found to be clustered according to species membership, the majority of the tetraploid accessions followed this pattern except representatives of *L. ircutianum* subsp. *ircutianum* and *L. delarbrei* subsp. *ruscinonense*. Regarding the former, the accessions were observed to form two independent clusters comprising two and five accessions (the latter including L055-03 and its replicate L055-031), and a single individual (437-01) was ungrouped with any cluster. For *L. delarbrei* subsp. *ruscinonense,* two clusters with two (accessions 139-01 and 355-03) and three (accessions 101-01, 110-01, and 349-01) individuals were observed in quite distant positions in the network. When subjecting the tetraploid accessions alone to Neighbor-net clustering (Figure 2B), sample 437-01 was found to be grouped with the larger of the two *L. ircutianum* subsp. *ircutianum* clusters. As a consequence, this cluster comprises accessions of the taxon from Austria (L062-04), Corsica (437-01), Germany (L052-02, L055-03), Italy (87-01), and Montenegro (177-01), while the second one, closer to *L. cantabricum*, contains the two accessions from SW France (106-01 and 343-01). Additionally, all accessions of *L. delarbrei* subsp. *ruscinonense* now cluster together, and the network reconstruction indicates that there are closer genetic similarities between (a) *L. delarbrei* subsp. *delarbrei* and *L. meridionale* (both S France) and (b) *L. crassifolium* (N Spain)*, L. pseudosylvaticum* (NW Spain), and *L. delarbrei* subsp. *ruscinonense* (SW France, NE Spain), while accessions of *L. ircutianum* subsp. *leucolepis* (Italy, Balkan Peninsula) form a cluster relatively isolated from these.

The optimal number of clusters according to the Bayesian information criterion (BIC; ‘lower is better’) was found to be six (Figure 3B), while the Calinksi–Harabasz (CH; ‘higher is better’) score was optimal for *k* = 2, but also had a local optimum at six clusters. The consensus clustering with the co-association matrix at *k* = 6 (Figure 3A) merged *L. meridionale* and *L. delarbrei* subsp. *delarbrei* (pink cluster) and *L. ircutianum* subsp. *ircutianum* and *L. cantabricum* (yellow cluster), while the remaining assignments of accessions to clusters followed their taxonomic classifications.

### 3.4. Genealogical Species Delimitation

Plots of the category proportions for all taxon triplets surveyed in the SNiPloid approach to infer parental diploids are provided in the supporting information (Appendix A). We found that SNP categories 1 and 2 increased with the genetic distances of the corresponding parental species from the child taxa. The sum of categories 3 and 4 generally decreased with genetic distance. For the analysed taxa, we could not find any irregularity in this pattern. The frequency of category 5 was relatively constant, but also slightly decreased with the genetic distances of the parental species. There seemed to be slight shifts in category frequencies, with at least one parent outside of the *L. vulgare* group (i.e., *L. vulgare*, *L. gaudinii*, *L. pluriflorum*, *L. ageratifolium*, *L. monspeliense*, *L. legraenum*, and *L. ligusticum*) included as a parental species.

## 4. Discussion

Having provided an integrative classification scheme for the diploid representatives of the genus [19], the present contribution aims at a comparable taxonomic treatment of tetraploid *Leucanthemum* taxa based on the same sources of evidence for species delimitation: genetic diversification, morphological discontinuities, ecological differentiation, and information on the geographical distribution of taxa. In addition to the methodological approach taken in the diploid case, we additionally tried to incorporate genealogical aspects into this integrative taxonomy of *Leucanthemum* tetraploids: by trying to infer the parentage of tetraploid taxa from RADseq-based single-nucleotide polymorphism (SNP) data, we hoped for additional arguments for a classification scheme that incorporates the evolutionary history of the study group. This follows the rationale that, by elucidating the combination of diploid genomes in the polyploids, the disentangling of auto- and alloploid formations of these taxa and the knowledge of their independent vs. non-independent evolutionary trajectories could be used in taxonomic decisions.

### 4.1. Genealogical and Genetic Patterns

Unfortunately, the present analyses provide us only with a quite limited notion about the evolutionary origins of the tetraploid *Leucanthemum* taxa under study. This disappointing result is somewhat unexpected in consideration of the huge amount of genomic information received from the GBS fingerprinting, with over 150,000 SNPs from over 7000 loci. However, our present analyses simultaneously point towards the probable explanation for the unresolved evolutionary patterns in this plant group: all eight surveyed tetraploid taxa appear to be closely related to the close-knit species group of diploids around *L. vulgare* (Figure 2A), for which already at the diploid level species delimitation and reconstruction of phylogenetic relationships were found to be problematic [19]. Owing to the fact that the radiation of the whole genus *Leucanthemum* was dated to the last 1.9 (1.1–2.9) Ma [27] and the diversification of the *L. vulgare* group is presumably not older than 400 ka, polyploid species formation from these diploid ancestors has to be assumed to have occurred not earlier than the Middle (Chibanian, 0.77–0.13 Ma) or Late Pleistocene (0.13–0.01 Ma) with its glaciation cycles of the Mindel, Elster, Riss, and Würm eras.

It is comprehensible, therefore, that our present efforts to trace the parenthood of diploid species for the formation of tetraploid taxa by application of the SNiPloid-based strategy proposed by Wagner et al. [51] for *Salix* polyploids revealed no patterns in *Leucanthemum*. We think that the observed failure of the mentioned strategy, in contrast to its successful application in *Salix* tetraploids, is not due to a lack of sufficient loci and sampled SNPs (23,393 loci and 320,010 SNPs in *Salix* vs. 7342 loci and 156,057 SNPs in *Leucanthemum*) but due to the fact that the putative parental diploid species in *Salix* are much older than in *Leucanthemum*. He et al. [52] provide a dated phylogeny for the *Salix* subg. *Chamaetia/Vetrix* clade, in which the two diploids *S. purpurea* L. and *S. repens* L.—the two putative parents of the tetraploid *S. caesia* Vill. analysed by [51] using SNiPloid—are dated as having diverged from each other in the Early Miocene, around 20 Ma ago. It appears obvious that in a constellation like this, with the long evolutionary independence of diploid precursor lineages and a relatively recent formation (Pleistocene) of a tetraploid taxon, the SNiPloid-based strategy proposed by Wagner et al. [51] will reveal a clear signal. However, this cannot be expected in the *Leucanthemum* case, with its comparatively young diploid lineages.

There may be methodological concerns that the SNiPloid approach is strongly influenced by the filtering of RADseq reads gained from polyploids. This is due to the fact that, when parameters (cluster thresholds) are optimised in iPyrad to avoid under- and oversplitting of loci and to minimise paralogy, paralogy caused by whole-genome duplication (i.e., homoeology) will be also reduced and this could diminish potential additive allelic signals of allopolyploidy. However, we circumvented this problem by applying a reference-based assembly of reads from tetraploids (mapped against the diploid reference from Ott et al. [19]) instead of performing a de novo assembly of reads from both diploid and tetraploid accessions. As a consequence, we are confident that our negative SNiPloid results are mainly due to the young age of polyploidisation events in *Leucanthemum*.

Nevertheless, despite the failure to pinpoint potential diploid parental taxa of the tetraploids using the SNiPloid-based strategy, some relationships among taxa at the two ploidy levels were revealed by the SNP-based network reconstruction (Figure 2). The most obvious connection of a tetraploid taxon with diploid species is the case of the NW Iberian *L. pseudosylvaticum,* which was found to cluster with the group comprising the N Spanish *L. eliasii* and the three subspecies of the NW Spanish *L. pluriflorum*. This position supports former reconstructions based on AFLP fingerprinting and sequence variation in cpDNA intergenic spacers and nrDNA ETSs [24,30] that all pointed towards the contribution of *L. pluriflorum* subsp. *pluriflorum* to an alleged allopolyploid origin of *L. pseudosylvaticum,* while they remained equivocal with respect to the donor of the other diploid genome to the latter. The allopolyploid nature of *L. pseudosylvaticum* receives considerable support from our present SNP-based analysis due to the obvious position of the three accessions of the species (1-05, 3-08, and 7-02) at the vertex of a parallelogram at their base in the NeighborNet network of Figure 2A. Following an interpretation of this parallelogram as an indication for *L. pseudosylvaticum* sharing one edge (genome) with *L. pluriflorum*/*L. eliasii,* one may feel justified in hypothesising the sharing of the other edge (genome) with one of the diploid species making up the left side of the network. Owing to the fact that all other members of this subgroup are presently not found in the Iberian Peninsula, *L. gracilicaule*—a diploid endemic to the region around Valencia in SE Spain—could then be the most probable candidate for the other parental taxon of *L. pseudosylvaticum,* if an extinct diploid may not have acted as such. The latter scenario receives plausibility from studies on other polyploid complexes that have encountered so-called ‘ghost (sub)genomes’ of extinct diploids in polyploid taxa (e.g., in *Viola* [53], in *Fragaria* [54], and in *Brachypodium* [55]).

In contrast to the evolutionary history of *L. pseudosylvaticum,* hypotheses about the formation of the other seven tetraploid taxa included in the present study remain even more obscure due to the nearly star-like structure in the right part of the NeighborNet network of Figure 2A. The position of the closely related diploid species *L. gaudinii* and *L. vulgare* (with its two subspecies *L. vulgare* subsp. *vulgare* and subsp. *pyrenaicum*) amongst accessions of the tetraploid taxa *L. ircutianum* (with its two subspecies *L. ircutianum* subsp. *ircutianum* and subsp. *leucolepis*) and *L. cantabricum* may point towards the participation of either of these two diploids (or a joint ancestor of both) in the formation of the latter. In the case of *L. ircutianum* subsp. *ircutianum,* it has been shown by Oberprieler et al. [28] that *L. vulgare* subsp. *vulgare* may have acted as the paternal partner in this allopolyploidisation and *L. virgatum* as the maternal partner [56]. At least the latter parentage receives little support from our present SNP-based reconstructions that locate *L. virgatum* quite distantly from all members of the *L. vulgare* cluster. Strangely enough, however, the cpDNA-based analysis of Greiner et al. [56] demonstrated that chloroplast haplotypes of *L. virgatum* (or haplotypes closely related to these) could not only be found in accessions of *L. ircutianum* subsp. *ircutianum* but also in some or even all representatives of *L. ircutianum* subsp. *leucolepis, L. cantabricum,* and *L. crassifolium* included in the mentioned study. Finally, chloroplast haplotypes characterised for *L. delarbrei* subsp. *delarbrei* and subsp. *ruscinonense* (sub *L. monspeliense*) were found to be closely related to *L. halleri* and *L. vulgare,* respectively [56]. In summary, these findings may either indicate real maternal parentages in the allopolyploidisation events (with subsequent assimilation of the nuclear genome towards the paternal parent) or events of chloroplast capture caused by hybridisation at the tetraploid level or between diploids and polyploids (tetraploids or taxa with even higher ploidy levels), indeed questioning the possibility of a comprehensive reconstruction of phylogenetic relationships among all taxa of the ploidy complex of *Leucanthemum*.

In genetic terms, SNP-based species-delimitation analyses carried out by weighted ensemble of random *(k)k*-means (WKM) clustering [49] and determination of the optimal number of clusters according to the Bayesian information criterion (BIC) and Calinski–Harbaz (CH) scores revealed significant discontinuities among six of the eight morphotaxon hypotheses (Figure 3). A lack of sufficient genetic differentiation was inferred for *L. cantabricum* and *L. ircutianum* subsp. *ircutianum* and for *L. meridionale* and *L. delarbrei* subsp. *delarbrei,* respectively. In both cases, this was found to correspond to the close positions of accessions from these taxon pairs in the NeighborNet network for the tetraploids (Figure 2B). In methodological respects, WKM clustering is a pattern-based, phenetic species-delimitation procedure not equivalent to process-based species-delimitation methods resting on the multispecies coalescent (MSC), which are used in many present-day delimitation studies based on molecular data in diploid taxon groups (e.g., [19,26,27] in *Leucanthemum* and [57] in *Rhodanthemum*). The lack of an MSC-based model adapted for polyploids hampers coalescent-based species delimitation here. However, since we have demonstrated in *Leucanthemum* diploids that pattern-based methods, such as consensus *k*-means (CKM; [58]) clustering, revealed genetic differentiation patterns equivalent to those derived according to a coalescent-based species-delimitation approach [19], we are confident that the differentiation patterns among tetraploids inferred with WKM clustering are trustworthy approximations to their genealogical structures.

It is obvious from both the NeighborNet network (Figure 2) and the WKM clustering (Figure 3) that *L. delarbrei* subsp. *delarbrei* and *L. meridionale* are closely related to each other. Despite the lack of significant genetic differentiation between them, as indicated by the BIC and CH scores, the two taxa appear to be genetically homogenous. This is remarkable due to the dispersed origins of accessions of the former taxon from four different locations on two volcanic mountain massifs in the southern Massif Central, France (Puy de Sancy, Monts du Cantal), while the latter is endemic to Puy de Wolf, a serpentine mountain c. 65 km SW of the Monts du Cantal. The situation is different for the other taxon pair, for which genetic differentiation was found to be not significant according to BIC and CH scores. In this case, *L. cantabricum*, with four accessions from two populations in NW Spain (Galicia), is also distinct but nested in a highly diverse cluster representing *L. ircutianum* subsp. *ircutianum*, with eight accessions from eight populations sampled throughout the European range of the taxon.

Two circumstances, however, are considered noteworthy here in evaluating the genetic/genealogical relationship of the two taxa: (a) The two populations sampled for *L. cantabricum* in the present analysis are the westernmost ones in the distributional range of the species that is mainly distributed in the W Pyrenean mountains and the Cordillera Cantábrica in N Spain [23], the *locus classicus* of the taxon being around 250 km to the east of the sampled populations. (b) In his revision of the genus for the Iberian Peninsula, Vogt [22] considered the stronger dissected leaves of *L. cantabricum* (sub *L. ircutianum* subsp. *cantabricum*) as being the main difference between this taxon and *L. ircutianum* subsp. *ircutianum*. The author, however, reports also that the morphological variation observed in the former taxon is considerable and that morphologically intermediate individuals with respect to the latter are observed frequently. Sampling of accessions from the centre of the distributional range of *L. cantabricum* (especially from the *locus classicus*) may reveal more pronounced genetic differences to *L. ircutianum* subsp. *ircutianum*, and the two populations sampled for the present study may turn out to be intermediate forms rather than pure representatives of *L. cantabricum*. Nevertheless, the undeniable closeness of the two taxa and their connectedness via intermediates will remain an unequivocal genealogical pattern.

### 4.2. Morphological Patterns

Leaf morphology—especially leaf outline and leaf dissection—is of paramount importance for taxon delimitation and taxon determination in *Leucanthemum*. As demonstrated by many taxonomic treatments of the genus in floras of European countries (e.g., *Flora Iberica* [23], *Flora Gallica* [59], and *Flora d´Italia* [60]), other morphological features, such as the colour of margins of involucral bracts, plant size, number of capitula per stalk, and achene size, come only second to these characters. Therefore, we felt justified in utilising leaf morphology as the sole proxy for morphological variation in the study group, especially because leaf-morphological features could be easily and objectively inferred using the applied machine-learning techniques and images of digitised herbarium specimens. These techniques allowed us to analyse 285 representative specimens and more than 1300 individual leaves across the study group in terms of leaf dissection and leaf shape in a time-effective and reproducible manner and to test for significant differences between pairs of the eight morphotaxa. We are fully aware of the fact that significant differences in these tests do not indicate the suitability of these characteristics as sole taxon-diagnostic features, as all taxa show considerable variation and overlap broadly in leaf characteristics. However, the main motivation for the inclusion of morphological characteristics in species-delimitation analyses is not for practical reasons of applicability (determinability), but—as in the case of genetic/genealogical features—for the assessment of morphological discontinuities as potential proxies for the evolutionary independence of lineages and their reciprocal reproductive isolation.

When compared across all possible pair-wise test combinations, leaf dissection (as measured by LDI) discriminates among the eight morphotaxa more profoundly than leaf outline (measured in elliptic Fourier analyses and tested for significance either with a permutation or an NPMANOVA approach): while only three (11%) of the 28 possible pair-wise comparisons of LDIs revealed no significant differences between taxa, 14 (50%) and 12 (43%) of the tests addressing leaf-shape differences showed non-significant results. We think that these contrasting results do not only have a biological reason, but also a methodological one: in its present and here-applied formulation (comparison of the length of leaf outline with leaf area), LDI does not only measure the intensity of leaf incision, but also the deviation of the leaf shape from a perfect circle. That means that it also captures leaf shape. In contrast, elliptic Fourier analysis (EFA) not only exclusively captures the outline of leaves, but also (with higher harmonics) the subdivision of the leaf lamina. However, since parameters gained with the latter harmonics of the EFA are down-weighted against parameters from earlier ones in a principal component analysis (PCA), it may be justified to consider both measures applied here (i.e., LDI and EFA) as overlapping with respect to what they capture in terms of leaf morphology, but with a tendency towards assessment of leaf dissection with the former and leaf outline with the latter.

Discrepancies between LDIs and EFA descriptors of leaf morphology were found to be especially pronounced when *L. meridionale* was included in the pair-wise comparisons: while six of the seven comparisons concerning LDIs revealed significance, the majority of EFA-based comparisons resulted in non-significant results. The former is quite reasonable when the habit of the species is studied both in its natural habitat and in a herbarium: compared with all other tetraploid taxa of *Leucanthemum, L. meridionale*, with its reduced scape size and its filigree leaves, resembles more the diploid representatives of the genus than the tetraploid ones. This led Tison and de Foucault [59] among others, who did not know about the tetraploid nature of this taxon, to speculate on the close (and allegedly) hybridogenic relationship of the species to *L. vulgare* and *L. graminifolium*. The somewhat reduced habit of *L. meridionale* may be a consequence of the exceptional habitat preference of the species for serpentine soils; the lack of a significant difference in LDIs with respect to *L. delarbrei* subsp. *delarbrei,* which is found on volcanic soils further north and is genetically closely related with it, however, may also point towards a close phylogenetic relationship between these two tetraploid *Leucanthemum* taxa.

A further noteworthy similarity in terms of leaf-morphological descriptors was observed between *L. ircutianum* subsp. *ircutianum* and subsp. *leucolepis,* for which all three tests remained non-significant (Table 1). As described by Oberprieler et al. [31], the former wide-spread taxon and the latter amphi-Adriatic one (including the S Italian *L. ircutianum* subsp. *asperulum* as a synonym) are allopatrically distributed in the Apennine Peninsula but sympatrically distributed in the Balkan Peninsula and form morphologically intermediate forms through hybridisation in overlapping regions of their distribution ranges. The main difference between the two taxa is in the colouration of the margins of involucral bracts, which are black to dark-brown in subsp. *ircutianum* and hyaline in subsp. *leucolepis*. In genetic respects, the two taxa represent two significantly distinct (Figure 3), albeit closely related (Figure 2), clusters.

### 4.3. Ecological and Geographical Patterns

As in the case of pair-wise statistical testing of leaf-morphological aspects, testing for significantly non-overlapping distribution ranges (Table 3) and ecological niches (Table 2) among the tetraploid *Leucanthemum* morphotaxa did not reveal complete and therefore diagnostic separation in these respects (strict allopatry or non-overlapping ecological niches); it rather led to the detection of patterns of bimodality in the spatial and ecoclimatological distributions of the species pairs compared. Therefore, the overwhelming number of significant test results (even after Bonferroni correction for multiple testing) for both geographical (2 of 21 tests) or ecological (also 2 of 21 tests) differences between species pairs may just represent tendencies in spatial or ecoclimatological differentiation and not complete discontinuities. Nevertheless, when looking at the distribution ranges of the eight tetraploid morphotaxa under study (Figure 1) and adding information from taxon descriptions concerning their ecological behaviour (e.g., [22] for the taxa of the Iberian Peninsula), it becomes obvious that the mentioned test results understate rather than hyperbolise differentiation patterns.

In geographical respects, the distribution ranges of the eight morphotaxa clearly follow an allopatric pattern, with exceptions being the observed non-significant differentiation between *L. ircutianum* subsp. *ircutianum* and *L. delarbrei* subsp. *ruscinonense* on the one hand and *L. cantabricum* and *L. crassifolium* on the other. As in the case of the point-endemic *L. meridionale,* for which statistical testing for geographical and ecological differences was not possible due to the restricted distribution range of the species, we think, however, that these test results are strongly scale-dependent and more on the conservative than on the oversensitive side. In the case of the taxon pair of *L. cantabricum* and *L. crassifolium,* the two species show a clear allopatric distribution on a small scale, with the former being restricted to the mountains of the Cordillera Cantábrica and adjacent regions [22,23] and the latter to littoral habitats of the N Spanish coast. In the other cases, the wide-spread (and constantly further spreading) *L. ircutianum* subsp. *ircutianum* encloses the geographically restricted *L. delarbrei* subsp. *ruscinonense* and *L. meridionale* in a more parapatric than allopatric pattern, which may be responsible for the non-significant test result. For all three taxon pairs, however, ecological differences were either revealed by our tests on niche overlap *(L. ircutianum* subsp. *ircutianum* vs. *L. delarbrei* subsp. *ruscinonense)* or by small-scale habitat differences—especially in edaphic factors—that have not been captured by the geography-based ecoclimatological niche modelling underlying the test-setup, *L. meridionale* being adapted to serpentine soils and *L. crassifolium* to saliferous coastal habitats.

### 4.4. Integration of Sources of Evidence

Species delimitation in plants—especially in hybridising plant groups (syngameons) and polyploid complexes—is a problematic matter [18]. The applicability of a strict biological species concept (BSC) has been denied by the majority of botanists owing to the sheer frequency of hybridisation in the plant kingdom and the numerous groups with agamospermic (apomictic) reproduction. Additionally, the usage of actual (hybridisation) or potential interbreeding (crossability) as a proxy for the evolutionary independence of lineages is deceptive. The lack of correlation between hybrid-formation capability and phylogenetic proximity in plants is exemplified by many examples of old and well-characterised lineages (species, sometimes even from different genera) that easily hybridise when brought into contact on the one hand and extremely young lineages (like autopolyploids) that are reproductively isolated instantly on the other, demonstrating that interfertility is a plesiomorphic character state [61]. Other species concepts, as enumerated by Zachos [62], share the problem that the criteria entertained in the different concepts differ in temporal sequence and relative importance across the tree of life due to the fact that speciation is a continuous process “over a timeframe that is too long to study from start to finish” (the ‘speciation continuum’; [63]). With these unsatisfactory consequences for biological classification, taxonomic decisions following a ‘unified species concept’ sensu De Queiroz [5] have considerable attraction because this concept shifts the focus away from properties being used for defining species towards their usage as indicators for independently evolving metapopulation lineages (species). Additionally, it allows for and demands the integration of multiple sources of evidence for taxonomic ranking.

Here, we follow a procedural protocol for species-rank decisions designed by Oberprieler [16] that is based on the integration of morphology, ecology, and geography, as proposed by von Wettstein [64], and expanded for the inclusion of an additional genetic/genealogical axis (the ‘Wettstein tesseract’). Conceptually, it follows the evolutionary species concept (EvoSC) of Wiley [65], who defined species as “a single lineage of ancestral-descendent populations of organisms which maintains its identity from other such lineages and which has its own evolutionary tendencies and historical fate”, but addresses the major drawback of that definition, namely, that genealogy-based, multispecies-coalescent species-delimitation methods tend to mistake population structures for species boundaries [66]. By adding geographical, ecological, and morphological sources of evidence, the ‘Wettstein tesseract’ provides a tool for conceptualising decisions on species or subspecific ranks and has been successfully applied to the diploids of *Leucanthemum*—the ”warps and wefts” of this polyploid complex [19].

Despite the somewhat pessimistic outlook on the phylogenetics of the *Leucanthemum* polyploidy complex, the present study is extremely helpful in terms of providing a pattern-based species delimitation at the tetraploid level and its integrative taxonomical conceptualisation. Aiming at a reproducible, objective classification of eight morphotaxon hypotheses in terms of delimitation and ranking, we have analysed genetic, ecological, and morphological variation together with information from taxon distributions, summarised in Figure 4. In the diagram, statistically significant differences found between pairs of taxon hypotheses are depicted in red, while the lack of significant discontinuities is shown in green. Owing to the restricted distribution range of *L. meridionale,* which is a point endemic of a serpentine mountain in S France (Aveyron, Rodez, Decazeville, Firmi, and Puy de Wolf), this taxon has been omitted from pair-wise comparisons that are dependent on modelling procedures based on spatial distribution data comprising more than a single raster cell (scaled to a resolution of 2.5 arc minutes). Due to its described endemicity, *L. meridionale* (accessions 323-03, -04, and -05) is allopatrically distributed with all other tetraploid *Leucanthemum* taxa except *L. ircutianum* subsp. *ircutianum,* which is found in parapatry on non-serpentine soils surrounding Puy de Wolf (and sampled with accession 343-01). Its ecological niche is uniquely determined by edaphic factors connected to serpentine soils (only paralleled by the diploid *L. pluriflourum* subsp. *gallaecicum* in NW Spain and the decaploid *L. pachyphyllum* in N Italy) rather than climatic ones.

Application of the ‘Wettstein tesseract’ tool to the present study group of *Leucanthemum* tetraploids in seeking the most reasonable ranking for closely related species hypotheses (morphotaxa) leads to the following reasoning:

(a) *Lecanthemum crassifolium* (N Spain), *L. delarbrei* subsp. *ruscinonense* (E Pyrenees and northeastern foothills)*,* and *L. pseudosylvaticum* (W Iberian Peninsula) are best considered as being genetically closely related but independent lineages (Figure 2 and Figure 3) that merit species ranking due to their significant ecological (Table 2) and leaf-morphological differences (Table 1) that have evolved in strict allopatry. While for *L. pseudosylvaticum* at least one parental diploid species is known (*L. pluriflorum* subsp. *pluriflorum*; [24,30]), the matter is unsettled as yet for the other two tetraploid lineages. Following haplotype-network reconstructions for the whole genus [56], *L. pluriflorum* subsp. *pluriflorum*, as the maternal diploid ancestor of these, could be excluded definitively, and *L. virgatum* appears to be the most probable candidate for this role in *L. crassifolium,* while *L. delarbrei* subsp. *ruscinonense* (sub *L. monspeliense* in [56]) shares its chloroplast haplotype with a large number of diploid taxa: *L. ageratifolium* (NE Spain, SW France; sub *L. vulgare* subsp. *pujiulae* in [56]), *L. burnatii* (S France), *L. eliasii* (N Spain; sub *L. vulgare* subsp. *eliasii* in [56]), *L. gaudinii* (Alps, Carpathian Mountains), *L. graminifolium* (S France), *L. pluriflorum* subsp. *cantabricum* (N Spain; sub *L. gaudinii* subsp. *cantabricum* in [56]), and both subspecies of *L. vulgare* (subsp. *vulgare,* widespread; subsp. *barrelieri,* Pyrenees).

(b) *Leucanthemum delarbrei* subsp. *delarbrei* (northern Massif Central, France) and *L. meridionale* (western Massif Central, France) are allopatrically distributed sister groups (Figure 2) that lack sufficient genetic differentiation to merit species ranking (Figure 3) and show non-significantly different leaf morphology (Table 1) but exhibit edaphic differences (the former growing on siliceous rocks of old volcano cones, the latter on serpentine soils). Therefore, subspecific ranking appears appropriate for these two taxa. The chloroplast haplotype found in one of two accessions of *L. delarbrei* subsp. *delarbrei* surveyed by Greiner et al. [56] was found to be closely related to the diploid *L. halleri* and may point towards a phylogenetic relationship with this Alpine species, while information on a chloroplast haplotype in *L. meridionale* is still lacking.

(c) Finally, *L. cantabricum* and the two subpecies of *L. ircutianum* also show close genetic relationships in the NeighborNet network of Figure 2. Here, however, genetic differentiation patterns (Figure 3) do not correspond to the hitherto proposed classification because *L. ircutianum* subsp. *ircutianum* shows a closer genetic similarity with *L. cantabricum* than with *L. ircutianum* subsp. *leucolepis*. When following the conceptual framework of species-rank decision making with the ‘Wettstein tesseract’, the significant differentiations between *L. ircutianum* subsp. *ircutianum* and subsp. *leucolepis* in genetic, geographical, and ecoclimatological aspects would argue for an acknowledgement of the two taxa at species level (the lack of leaf-morphological differences is counterbalanced by differences in the colours of the margins of the involucral bracts). The main argument for treating the two entities as independent species, following the logic of von Wettstein [64], is that the ecoclimatiological differences between the two would allow them to keep their lineage identity when allopatry would change into sympatry in the future. Oberprieler et al. [31] have shown, however, that there are mixed stands of the two taxa in Central Italy, where hybridisation and backcrossing leads to introgressive hybrid swarms, with a complete blurring of taxon limits. Since the situation seems comparable in the Western Balkan Peninsula, where the two taxa grow sympatrically and intermediate forms are found [31], we propose to keep the two entities at subspecific rank for conservative reasons aiming at minimizing taxonomic and nomenclatural disruptions if not unequivocally demanded by the underlying data and analyses.

Subspecific ranking under *L. ircutianum,* on the other hand, is less equivocal for *L. cantabricum* (Cordillera Cantábrica in N Spain) following the here-presented results: the two taxa are allopatrically distributed and show ecoclimatological and morphological differences (more strongly dissected leaves in *L. cantabrica*) but lack genetic differentiation (Figure 4). The observation of morphologically intermediates by Vogt [22] further argues for hybridisation between the two taxa and the appropriateness of subspecific ranking.

## 5. Taxonomic Treatment

With the described conceptual framework at hand, species delimitation in the group of tetraploid *Leucanthemum* taxa under study could be put into effect as follows:

(1) ***Leucanthemum crassifolium*** (Lange) Lange in Willk. & Lange, Prodr. Fl. Hispan. 2: 96. 1865 ≡*Leucanthemum pallens* var. *crassifolium* Lange in Vidensk. Meddel. Dansk Naturhist. Foren. Kjøbenhavn, ser. 2, 3: 77. 1861 ≡*Leucanthemum ircutianum* subsp. *crassifolium* (Lange) Vogt in Ruizia 10: 127. 1991—Lectotype (Vogt, Ruizia 10: 128. 1991): In rupibus maritimis ad Portugalete, Cantabria, Oct. 1851, Herb. Joh. Lange (C! (C10007128)).

*Notes*.—Endemic to the Cantabrian coast between Asturias and the Basque region (NE Spain and SW France). Habitats are coastal rocks and salt-influenced coastal slopes from sea level to 20 m. *Leucanthemum crassifolium* is characterised by succulent leaves (Figure 5) and involucral bracts with dark-brown hyaline margins of involucral bracts. It was first described as a variety of *L. pallens* by Lange [67] and subsequently raised to subspecific [22] and specific rank [23].

(2a) ***Leucanthemum delarbrei*** Timb.-Lagr. (**subsp. *delarbrei***) in Mém. Acad. Sci. Clermont-Ferrand 20: 508. 1878.—Lectotype: Not yet designated. =*Chrysanthemum leucanthemum* var. *pinnatifidum* Lecoq. & Lamotte, Cat. Pl. Plateau Central: 227. 1847 ≡*Leucanthemum vulgare* var. *pinnatifidum* (Lecoq. & Lamotte) Briq. & Cavill. in Burnat, Fl. Alpes Marit. 6: 91. 1916 ≡*Leucanthemum ircutianum* var. *pinnatifidum* (Lecoq. & Lamotte) D. Löve & J.-P. Bernard in Svensk. Bot. Tidskr. 53: 444. 1959.—Ind. loc.: “AR.- Mont Dore; pâturages et pentes herbeuses de Chaudefour, bords du chemin de Sancy à Vassivière. Bozat! AR.”—**Lectotype (designated here by Vogt & Oberprieler):** Mont Dore, Sancy, 15 (..) 1844, leg. *H. Lecoq & M. Lamotte* (P! (P00729975)).

*Notes*.—Endemic to the siliceous rocks of the old volcano cones of the central Massif Central in France (Monts Dore, Monts du Cantal). Its habitats are siliceous rocks and meadows between 1550 m and 1750 m. *Leucanthemum delarbrei* subsp. *delarbrei* is characterised by strongly dissected leaves, from pinnatifid to bipinnatisect (Figure 5), and dark- to light-brown hyaline margins of involucral bracts.

(2b) ***Leucanthemum delarbrei* subsp. *meridionale*** (Legrand) Oberpr., T. Ott & Vogt, **comb. nov.** ≡*Leucanthemum meridionale* Legrand in Bull. Soc. Bot. France 28: 56. 1881 (basionym) ≡*Leucanthemum vulgare* var. (‘χ’) *meridionale* (Legrand) Rouy, Fl. France 8: 274. 1903 ≡*Chrysanthemum leucanthemum* f. (‘g’) *meridionale* (Legrand) Fiori in Fiori & Béguinot, Fl. Italia 3: 239. 1903 ≡*Leucanthemum vulgare* subsp. *meridionale* (Legrand) Nyman, Consp. Fl. Eur., Suppl. 2: 169. 1889.—Ind loc.: “Habite dans les interstices des rochers serpentineuses du puy de Wolf, près de Firmy (Aveyron); fleurit de fin mai à juillet. Je reçus cette plante en 1879 ... leg. *F(rère) Saltel* (Baenitz, Herbarium eur. n° 4184).”—**Lectotype (designated here by Vogt & Oberprieler):** Puy de Wolf, pr. Firmy; mai, juin 1879 (Aveyron).—Gallia merid., Saltel.—Comm. Le Grand (P! (P00729957)).

*Notes.*—Endemic to the Puy de Wolf (France, Aveyron, Rodez, Decazeville, Firmi), a serpentine mountain in the western Massif Central. It is found in dry and open, south-facing grassland on serpentine soil between 400 m and 600 m. *Leucanthemum meridionale* is characterised by its lanky habitus resembling the diploid *L. vulgare* or a member of the genus *Leucanthemopsis* with quite narrow leaves and cuneate leaf bases (Figure 5).

(3a) ***Leucanthemum ircutianum*** DC. (**subsp. *ircutianum***), Prodr. 6: 47. 1838.—Lectotype (Vogt, Ruizia 10: 119. 1991): In pratis, 1828, Turczaninoff a Irkoutsk, Turcz.: 1830 (G-DC! (G00451151)).

*Notes*.—Besides the diploid *L. vulgare* Lam., this taxon is the most widely distributed species of the genus [22]. It is present in nearly all countries of the Euro + Med region [20] but has also been introduced into all continents except Antarctica. Habitats are anthropogenetically influenced and include meadows and roadsides. *Leucanthemum ircutianum* subsp. *ircutianum* is morphologically very similar to *L. ircutianum* subsp. *leucolepis* (see Figure 5) but can be differentiated by the dark-brown hyaline margins of involucral bracts.

(3b) ***Leucanthemum ircutianum* subsp. *cantabricum*** (Sennen) Vogt in Ruizia 10: 121. 1991 ≡*Leucanthemum cantabricum* Sennen, Diagn. Nouv.: 50. 1936 ≡*Leucanthemum vulgare* subsp. *cantabricum* Sennen, Diagn. Nouv. 50. 1936 (nom. altern.).—Lectotype (Vogt, Ruizia 10: 121. 1991): Santander: La Calda de Besaya, rochers silliceux humides, 5.6.1927, *E. Leroy* (BC-Sennen!).

*Notes*.—Distributed between the western foothills of the Pyrenees (Spain and France), along the northern slopes of the Cantabrian Mountains to Asturias and Galicia in the west. Habitats include road and meadow margins, meadows, and pastures; the elevational distribution ranges from sea level to 800 m. *Leucanthemum cantabricum* is characterised by dissected (pinnatisect to pinnatipartite), non-succulent leaves (Figure 5) and dark-brown hyaline margins of involucral bracts. It was described as a species by Sennen [68] but reduced to subspecific rank under *L. ircutianum* by Vogt [22] before being reacknowledged at species rank by Vogt [23].

(3c) ***Leucanthemum ircutianum* subsp. *leucolepis*** (Briq. & Cavill.) Vogt & Greuter in Willdenowia 33: 41. 2003 ≡*Leucanthemum leucolepis* (Briq. & Cavill.) Gajić in Josifović, Fl. SR Srbije 9: 185. 1977 ≡*Leucanthemum vulgare* subsp. *leucolepis* Briq. & Cavill. in Burnat, Fl. Alpes Marit. 6: 93. 1916 ≡*Chrysanthemum leucanthemum* subsp. *leucolepis* (Briq. & Cavill.) Schinz & Thell., Fl. Schweiz, ed. 4, 1: 685. 1923 ≡*Leucanthemum leucolepis* (Briq. & Cavill.) Horvatić in Acta Bot. Croat. 22: 214. 1963, nom. inval. (Turland et al. 2018: Art. 41.5) ≡*Leucanthemum pallens* subsp. *leucolepis* (Briq. & Cavill.) Faverger in Anales Inst. Bot. Cavanilles 32: 1236. 1975.—Lectotype (Oberprieler et al., Bot. J. Linn. Soc. 199: 844. 2022): Flora Italica Exsiccata, curantibus Adr. Fiori, A. Béguinot, R. Pampanini, number 175—Etruria, Prov. di Firenze, Vallombrosa, in pratis, alt. 900–1000 m., solo pingui, siliceo, 19.6.1904, *A. Fiori* (G-BU! (G00848032)).

*Notes.*—A taxon with an amphi-Adriatic distribution, with populations throughout Italy and along the Adriatic coast of the Balkan Peninsula. The taxon was described as a subspecies of the diploid *L. vulgare* but was subsequently considered a subspecies of the hexaploid *L. pallens* due to its white hyaline margins of involucral bracts or seen as an independent species. The observation of an allopatric distribution with *L. ircutianum* subsp. *ircutianum* as more Mediterranean facies of the species, together with the observation of hybrid individuals and hybrid swarms in areas of joint occurrence of the two tetraploid taxa in the Apennine and Balkan Peninsulas by Oberprieler et al. [31] argued for treatment as a subspecies of *L. ircutianum*. In contrast to *L. ircutianum* subsp. *ircutianum*, it has paler hyaline margins of the involucral bracts.

(4) ***Leucanthemum pseudosylvaticum*** (Vogt) Vogt & Oberpr. in Ann. Bot. (Oxford), n.s., 111: 1121. 2013 ≡*Leucanthemum ircutianum* subsp. *pseudosylvaticum* Vogt in Ruizia 10: 134. 1991.—Holotype: Portugal, Distrito Porto, Serra do Marão, Mesão Frio-Amarante, feuchter Hang südlich der Paßhöhe, ca. 800 m, 20.07.1986, *R. Vogt* 4711 & *E. Bayón* (M! (M-0030173)).

*Notes.*—Distributed throughout the western part of the Iberian Peninsula (Portugal and Spain). Its habitats comprise road margins and slopes, margins of creeks, and ditches, at an elevational range from 100 m to 1600 m. *Leucanthemum pseudosylvaticum* is characterised by leaves with proximally (sub)entire margins (Figure 5) and involucral bracts with pallid to light-brown hyaline margins. It was described by Vogt [22] as a subspecies of *L. ircutianum* but shown to merit species rank due to its evolutionary independence from the latter species by Oberprieler et al. [24] and Greiner et al. [29,30].

(5) ***Leucanthemum ruscinonense*** (Jeanb. & Timb.-Lagr.) Oberpr., T.Ott & Vogt, **comb. et stat. nov.** ≡*Leucanthemum palmatum* var. *ruscinonense* Jeanb. & Timb.-Lagr. in Mém. Acad. Sci. Toulouse, ser 8, 1(2): 192. 1879 (basionym) ≡*Leucanthemum monspeliense* var. *ruscinonense* (Jeanb. & Timb.-Lagr.) O. Bolòs & Vigo in Collect. Bot. (Barcelona) 17: 91. 1988 ≡*Leucanthemum cebennense* var. *ruscinonense* (Jeanb. & Timb.-Lagr.) Gaut., Catal. Rais. Fl. Pyr. Orient.: 232. 1898 ≡*Leucanthemum delarbrei* subsp. *ruscinonense* (Jeanb. & Timb.-Lagr.) Vogt, Florian Wagner & Oberpr., Fl. Iber. 16(3): 1866. 2019—Ind loc.: “... les Alberes ...”.—Holotype: ... la tour de la Massane dans la altura, Pyr. Orient., 21.5.1877, *E. Timbal-Lagrave* (TL-Timbal-Lagrave!).

*Notes.*—Endemic to the eastern Pyrenees (Spain and France) and the SW parts of the Massif Central (Haut Languedoc, Montagne Noire). Habitats comprise road margins, creeks, and stony slopes, between 250 m and 1900 m. It is characterised by strongly dissected leaves, pinnatifid to bipinnatisect (Figure 5), and dark- to light-brown hyaline margins. High variability in terms of leaf dissection is speculated to be the result of potential hybridisation. The taxon was, for a long time, considered part of *L. monspeliense* L. (e.g., [22]), which is also distributed in the Massif Central (Cevennes), but has a diploid chromosome number.

## Figures and Tables

**Figure 1 biology-12-00288-f001:**
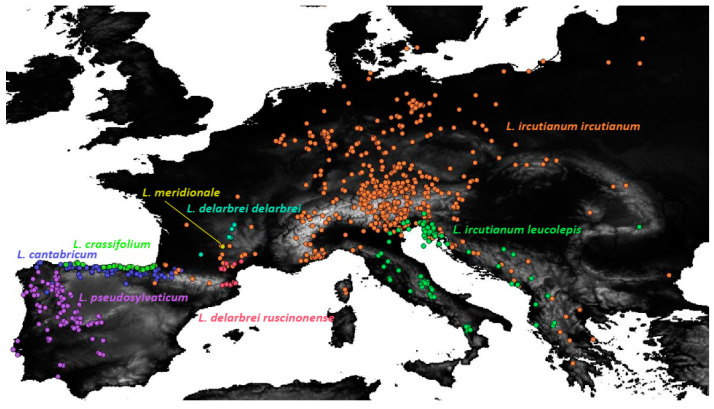
Distribution of the eight tetraploid *Leucanthemum* morphotaxa in Southern and Central Europe based on locality information from revised herbarium specimens which was used in the ecological niche modelling and geographical overlap analyses of the present study (see Appendix A for the list of georeferenced accessions).

**Figure 2 biology-12-00288-f002:**
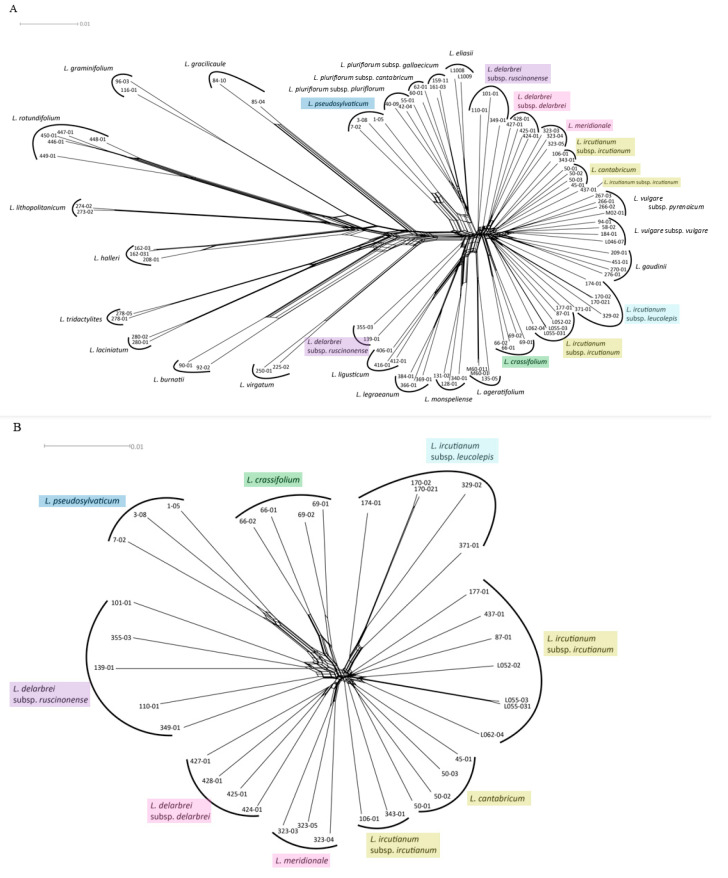
NeighborNet network reconstructions based on SNP-level Nei distances of the full dataset (**A**), comprising individuals from all 17 diploid species and eight tetraploid taxa (in coloured boxes), and of the tetraploids only (**B**). For the full dataset, all tetraploid taxa are found within the so-called *L. vulgare*-group (right). The *L. vulgare-*group is separated from the remaining diploid species by relatively long branches (center). All diploid samples are clustered according to their species membership (**A**). The same is the case for the tetraploid taxa, except for *L. ircutianum* subsp. *ircutianum*, which is separated into two groups ((**B**), lower right).

**Figure 3 biology-12-00288-f003:**
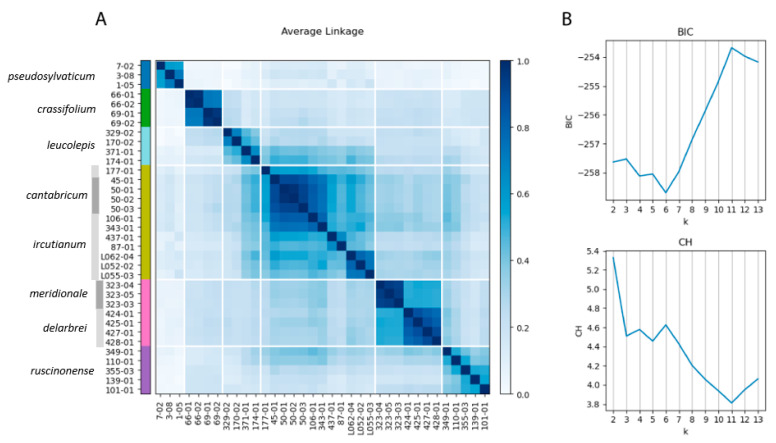
(**A**) Co-association matrix for the tetraploid *Leucanthemum* taxa based on weighted ensemble of random *(k)k*-Means (WKM) clustering and SNP-based Nei distances. Consensus clusters for *k* = 6 are shown, with colour bars depicting consensus cluster membership. (**B**) BIC and CH metrics for number of clusters ranging from *k* = 2 to *k* = 13.

**Figure 4 biology-12-00288-f004:**
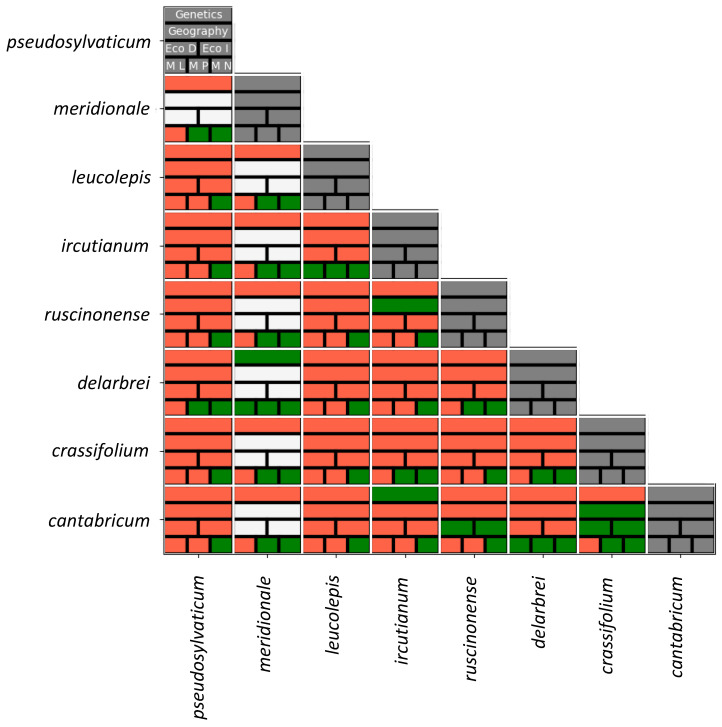
Taxon pair-wise differences in genealogy, geography, ecology (EcoD: Schoener´s D; EcoI: Warren´s I), and morphology [ML: Welch’s test on leaf-dissection indices (LDI); MP: permutation test on leaf shape (Elliptic Fourier Analysis, EFA), MN: NPMANOVA on leaf shape (EFA)]. Significant differences are marked in red, non-significant ones in green; white cells indicate lack of testability due to the restricted geographical range of *L. meridionale*.

**Figure 5 biology-12-00288-f005:**
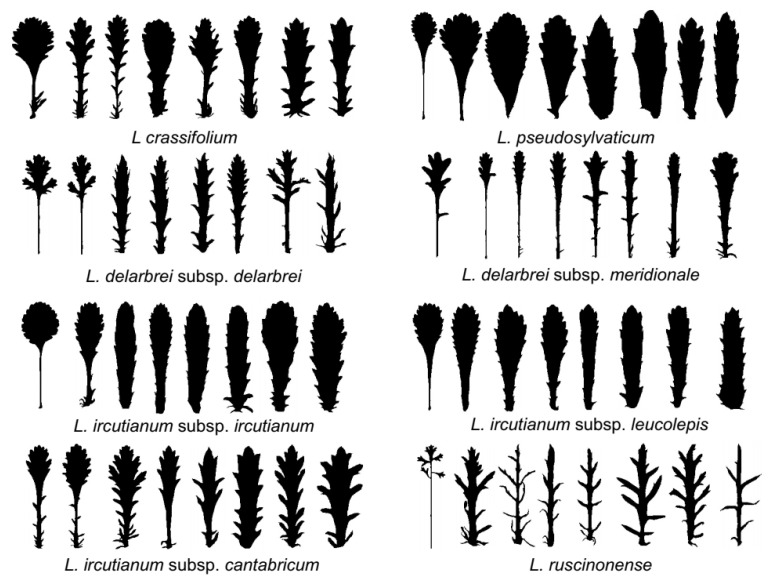
Silhouettes of basal and cauline leaves from different accessions of the eight tetraploid *Leucanthemum* taxa under study.

**Table 1 biology-12-00288-t001:** Bonferroni-corrected *p*-values for pairwise tests of morphological similarity. The upper triangle represents *p*-values of the LDI-based Welch’s tests, determining divergence in the dissection of the leaves. The lower triangle comprises *p*-values of the permutation tests of Euclidean distances in PC space (left) and NPMANOVA results (right), quantifying differences in general leaf shape. Corrected *p*-values are truncated to 1.0.

	*cantabricum*	*crassifolium*	*delarbrei*	*ruscinonense*	*ircutianum*	*leucolepis*	*meridionale*	*pseudosylvaticum*
** *cantabricum* **		<0.01	0.21	<0.01	<0.01	<0.01	<0.01	<0.01
** *crassifolium* **	1.0/1.0		<0.01	<0.01	<0.01	<0.01	<0.01	<0.01
** *delarbrei* **	1.0/1.0	0.44/1.0		<0.01	<0.01	<0.01	1.0	<0.01
** *ruscinonense* **	0.04/<0.01	0.02/<0.01	1.0/1.0		<0.01	<0.01	<0.01	<0.01
** *ircutianum* **	<0.01/<0.01	0.17/<0.01	0.02/0.83	<0.01/<0.01		1.0	<0.01	<0.01
** *leucolepis* **	<0.01/<0.01	<0.01/<0.01	<0.01/<0.01	<0.01/<0.01	1.0/<0.01		<0.01	<0.01
** *meridionale* **	1.0/1.0	1.0/1.0	1.0/0.83	1.0/1.0	1.0/<0.01	0.38/<0.01		<0.01
** *pseudosylvaticum* **	<0.01/<0.01	<0.01/<0.01	0.06/<0.01	0.01/<0.01	<0.01/<0.01	<0.01/<0.01	1.0/1.0	

**Table 2 biology-12-00288-t002:** Bonferroni-corrected *p*-values for pairwise niche equivalency tests based on the two test statistics, D (upper triangle) and I (lower triangle).

	*cantabricum*	*crassifolium*	*delarbrei*	*ruscinonense*	*ircutianum*	*leucolepis*	*meridionale*	*pseudosylvaticum*
** *cantabricum* **		0.52	0	0.32	0	0	n/a	0
** *crassifolium* **	0.31		0	0	0	0	n/a	0
** *delarbrei* **	0	0		0	0	0	n/a	0
** *ruscinonense* **	0.52	0	0		0	0	n/a	0
** *ircutianum* **	0	0	0	0		0	n/a	0
** *leucolepis* **	0	0	0	0	0		n/a	0
** *meridionale* **	n/a	n/a	n/a	n/a	n/a	n/a		n/a
** *pseudosylvaticum* **	0	0	0	0	0	0	n/a	

**Table 3 biology-12-00288-t003:** Bonferroni-corrected *p*-values for pairwise tests of geographical overlap, i.e., sympatry. Corrected *p*-values are truncated to 1.0.

	*cantabricum*	*crassifolium*	*delarbrei*	*ruscinonense*	*ircutianum*	*leucolepis*	*meridionale*	*pseudosylvaticum*
** *cantabricum* **		1.0	0	0	0	0	n/a	0
** *crassifolium* **			0	0	0	0	n/a	0
** *delarbrei* **				0	0	0	n/a	0
** *ruscinonense* **					0.05	0	n/a	0
** *ircutianum* **						0	n/a	0
** *leucolepis* **							n/a	0
** *meridionale* **								n/a

## Data Availability

The raw ddRAD reads were deposited at NCBI (Bioproject PRJNA884628).

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
