# Peer review of "Picks in the Fabric of a Polyploidy Complex: Integrative Species Delimitation in the Tetraploid Leucanthemum Mill. (Compositae, Anthemideae) Representatives"

_biology, 2023, doi:10.3390/biology12020288_

Round 1

Reviewer 1 Report

Having addressed the systematics and phylogeny of the diploid representatives of Leucanthemum in a previous paper, the authors deal in this manuscript with a tougher part of this genus, the polyploids. The team holds a leading position in integrative systematics of Anthemideae and, more in general, have experience in facing complex groups where they have made efforts to find suitable methodological approaches and shed light on scenarios involving hybridization and polyploidy.

This is a thoughtful and well-worked out piece of research. The results are honestly presented, highlighting the difficulties of finely reconstructing evolutionary history of polyploids and its representation in a classification. Given the importance of polyploidy in vascular plants, the methodological approaches followed and results obtained can be of interest for researchers beyond synantherology. The ms. is well-written in general although quite a number of sentences in the discussion are too long and a bit convolute. I thus think the ms. deserves publication.

Please find below a number of comments on specific points, some of them very minor, made for the consideration of the authors in the spirit of helping them improve the ms.

36-82_Understandably, in these paragraphs the authors make the very clear point that automatically generating a taxonomic treatment in plant groups that include polyploidy is almost a chimera. This is a proper introduction for what they present over the ms. This said, I would tone down a bit the relevance attributed to De Queiroz`s stand on species concept because the way it is presented, it seems that his view solves every problem and I don’t think this is case. But this is just my personal opinion.

45-48_This is a non-easy digestible sentence, particularly the last clause, which is rather convolute and could be formulated in simpler way.

93_”for discriminating taxa” seems quite out of place in that long sentence.

164_have been demonstrated in the diploids … and were assessed here too…

228-230_I see this procedure has been developed by the team in a previous paper. This brief mention, without specifying the magnitude of geographic gaps, sounds a bit bold in the sense that it seems to discard or preclude vicariance and long distance dispersal.

240-245_It is fine that they assess quality by re-extracting samples. I may have miss it, but I found no further comments on the results of these tests.

Table 1_Please check M60-01 and M60-011. The coordinates seem to be in the Cuenca province. I think “Teruel” should be removed.

Table 1_In 40-99 and 42-04, please replace “Corunna” by “Coruña”.

257 (also elsewhere in the main text)_”SNP-level” sounds strange. “SNP-based Nei distances” would be much better.  

395_Here and in a number of other sentences, the authors say “found being” instead of “found to be” (L. 442, 503,….)

408-416_I suggest that you make some statement here that addresses directly the heading of this part (Genealogical species delimitation) even if you develop it in the discussion. In the discussion (L.448) you jump to explaining why SNIPLOID failed. But you have not stated this previously in the results nor mentioned the program there.

434-485_Arguments to explain the poor results obtained in tracing the parentage of allopolyploids are fine. But it would be relevant to provide details and briefly discuss criteria applied when filtering the GBS reads. IPyrad and other programs tend to just minimize paralogy caused by whole genome duplication at the cost of eliminating potential signal. When dealing with a whole group of polyploid taxa, and given the poor results, some thoughts on this could be appropriate.

440_”close-knit” not closely-knit.

517_Figure 4, not 3.

551-556_Too long sentence, with the clause starting in “; even despite the origin…” unconnected to the previous. Please rephrase.

562_”judging”? Perhaps “exploring” or “analyzing”

572-577_very long sentence.

599-605_very long sentence.

616-630_It would be appropriate to discuss the single population of a polyploid (L. meridionale) in serpentine soils in a wider context (i.e. beyond Leucanthem). The tolerance or even adaptation of polyploids to serpentine substrates or others containing high proportions of heavy-metals is a recurrent topic.

644 (Fig. 4 caption)_I suggest that you give details of the symbols on top of the figure (EcoD EcoI, etc) as well as (in order to be used in the absence of the text) some allusion to the cause for white boxes that correspond to pairwise comparison involving L. meridionale.

648 ff_It is not clear where (figure, table) do the reader has to look to understand these statements.

682_syngamea might be a more correct plural than syngameons. But nobody uses it.

686_ “the inconsistent suitability of crossability or hybridisation as proxy for evolutionary independence”. Please rephrase into a simpler clause.

698-702_complex sentence. It was explained in a simpler way in the introduction.

706_”Conceptionally”? Do you mean “conceptually”?

Taxonomic treatment (L.769 ff)

Based on the taxonomic importance of leaf morphology, a figure displaying variation or representative leaf silhouettes for the tetraploids would be appropriate.

Author Response

Reviewer #1:

Having addressed the systematics and phylogeny of the diploid representatives of Leucanthemum in a previous paper, the authors deal in this manuscript with a tougher part of this genus, the polyploids. The team holds a leading position in integrative systematics of Anthemideae and, more in general, have experience in facing complex groups where they have made efforts to find suitable methodological approaches and shed light on scenarios involving hybridization and polyploidy.

This is a thoughtful and well-worked out piece of research. The results are honestly presented, highlighting the difficulties of finely reconstructing evolutionary history of polyploids and its representation in a classification. Given the importance of polyploidy in vascular plants, the methodological approaches followed and results obtained can be of interest for researchers beyond synantherology. The ms. is well-written in general although quite a number of sentences in the discussion are too long and a bit convolute. I thus think the ms. deserves publication.

Please find below a number of comments on specific points, some of them very minor, made for the consideration of the authors in the spirit of helping them improve the ms.

36-82_Understandably, in these paragraphs the authors make the very clear point that automatically generating a taxonomic treatment in plant groups that include polyploidy is almost a chimera. This is a proper introduction for what they present over the ms. This said, I would tone down a bit the relevance attributed to De Queiroz`s stand on species concept because the way it is presented, it seems that his view solves every problem and I don’t think this is case. But this is just my personal opinion.

No change here; we consider our statements appropriate. De Queiroz´ ‘species concept’ is best considered as being no concept (or species definition) at all, but a suggestion or tool to recognise independently evolving lineages and distinguish them from each other.

 45-48_This is a non-easy digestible sentence, particularly the last clause, which is rather convolute and could be formulated in simpler way.

Done; we have split the sentence into two for better palatability.

93_”for discriminating taxa” seems quite out of place in that long sentence.

Done.

164_have been demonstrated in the diploids … and were assessed here too…

Done.

228-230_I see this procedure has been developed by the team in a previous paper. This brief mention, without specifying the magnitude of geographic gaps, sounds a bit bold in the sense that it seems to discard or preclude vicariance and long distance dispersal.

Done; we have added “and unsampled” to be more precise here, but without describing the whole procedure again as described in the cited paper. Only unconnected areas containing no collected locality are removed here; this procedure does not exclude disjunct areas in the distribution range of a taxon in general, it only excludes disjunct areas that are not supported by any collection in that area.

240-245_It is fine that they assess quality by re-extracting samples. I may have miss it, but I found no further comments on the results of these tests.

Done; we have followed the following section to the Results section of the paper: “The trustworthiness of the RADseq fingerprinting procedure has been confirmed by the high degree of similarity between re-extracted and re-analysed accessions (L055-031, 170-021) and their counterparts (L055-03, 170-02) in the subsequent analyses.”

Table 1_Please check M60-01 and M60-011. The coordinates seem to be in the Cuenca province. I think “Teruel” should be removed.

Done; additionally, we have transferred Table 1 to the Supplementary Material (File ES04).

Table 1_In 40-99 and 42-04, please replace “Corunna” by “Coruña”.

Done.

257 (also elsewhere in the main text)_”SNP-level” sounds strange. “SNP-based Nei distances” would be much better. 

Done throughout the text.

395_Here and in a number of other sentences, the authors say “found being” instead of “found to be” (L. 442, 503,….)

Done; changed throughout the paper.

408-416_I suggest that you make some statement here that addresses directly the heading of this part (Genealogical species delimitation) even if you develop it in the discussion. In the discussion (L.448) you jump to explaining why SNIPLOID failed. But you have not stated this previously in the results nor mentioned the program there.

Done; we have changed the structure of the paragraph in the Results section (and added mentioning of the SNiPloid approach here) in order to address this issue.

434-485_Arguments to explain the poor results obtained in tracing the parentage of allopolyploids are fine. But it would be relevant to provide details and briefly discuss criteria applied when filtering the GBS reads. IPyrad and other programs tend to just minimize paralogy caused by whole genome duplication at the cost of eliminating potential signal. When dealing with a whole group of polyploid taxa, and given the poor results, some thoughts on this could be appropriate.

Done; we have addressed this issue by including a new paragraph in section 4.1. In contrast to the reviewer´s view we think that we have circumvented the mentioned GBS filtering issue by rather doing a reference-based assembly of reads from tetraploid accessions against a diploid reference than doing a de-novo assembly of reads from both diploids and tetraploids.

440_”close-knit” not closely-knit.

Done.

517_Figure 4, not 3.

Done.

551-556_Too long sentence, with the clause starting in “; even despite the origin…” unconnected to the previous. Please rephrase.

Done; we have split the sentence into two and slightly rephrased it.

562_”judging”? Perhaps “exploring” or “analyzing”

Done; changes to read “evaluating”.

572-577_very long sentence.

Done; we have split the sentence into two.

599-605_very long sentence.

Done; changed (but only slightly) by removing some words.

616-630_It would be appropriate to discuss the single population of a polyploid (L. meridionale) in serpentine soils in a wider context (i.e. beyond Leucanthemum). The tolerance or even adaptation of polyploids to serpentine substrates or others containing high proportions of heavy-metals is a recurrent topic.

Not done; owing to the length of the paper and its main focus on integrative taxonomy, we have omitted a broader discussion on serpentine endemics in general.

644 (Fig. 4 caption)_I suggest that you give details of the symbols on top of the figure (EcoD EcoI, etc) as well as (in order to be used in the absence of the text) some allusion to the cause for white boxes that correspond to pairwise comparison involving L. meridionale.

Done; we have addressed both point in the new caption of the figure.

648 ff_It is not clear where (figure, table) do the reader has to look to understand these statements.

Done; reference to tables 3 and 4 is given now in the text.

682_syngamea might be a more correct plural than syngameons. But nobody uses it.

Done; changed to read “syngameons”.

686_ “the inconsistent suitability of crossability or hybridisation as proxy for evolutionary independence”. Please rephrase into a simpler clause.

Done; we have rephrased the sentence.

698-702_complex sentence. It was explained in a simpler way in the introduction.

Done; we have changed the wording and split the sentence into two.

706_”Conceptionally”? Do you mean “conceptually”?

Done.

Taxonomic treatment (L.769 ff)

Based on the taxonomic importance of leaf morphology, a figure displaying variation or representative leaf silhouettes for the tetraploids would be appropriate.

Done; we have included a figure with leaf silhouettes of all eight taxa (Figure 5).

Reviewer 2 Report

This is an excellent manuscript addressing species delimitation in a polyploid complex. It follows an integrative taxonomic approach, exploring genomic, morphological, ecological and geographic  differentiation patterns, which are assessed here by several kinds of data analyses and tests, including state-of-the-art computations.

I have a few minor remarks that may aid in comprehensibility of the paper.

1)      The paper is extraordinarily long. I understand that is it methodologically complex, but the authors could consider some reduction of the Introduction and Discussion, to focus them on the most significant aspects and outcomes.

2)      I understand the format style of the references in the text adopted by the journal, which uses the numbers instead of the author names, but please try to avoid sentences such as: Finally, [27] presented….Most recently [19] combined….[22] considers…
It is so strange and disturbing reading such sentences.

3)      section 2.2: please include a picture illustrating the leaf shape variation or leaf shape analysis

4)      Table 1: It is too large and disrupts the continuity of the text. It would be more appropriate to place it as an appendix at the end of the paper.

5)      section 3.4: it would be nice and illustrative to include a graph or table here.

6)      Fig. 3: please specify that the analysis is based on SNP variation. Include also taxa names not just the accession labels in the (A) graph.

7)      Discussion, line 517:  Figure 4 should be referred to here instead of Figure 3

8)      Discussion, lines 511-529. This appears somewhat misplaced here, in the section ‘genealogical and genetic patterns’. Maybe it could be moved or incorporated in the section ‘4.4. Integration of sources of evidence’

9)      Discussion, lines 596-597. I completely understand and agree that the analyses aimed at finding morphological discontinuities, but I doubt these (especially regarding the leaf shape) can be considered as proxies for potential reproductive isolation.

10)   Figure 4: please explain abbreviations in the figure legend (Eco D, Eco I, ML, MP, MN)

11)   Discussion, lines 734-742. Please, specify here (like in the paragraphs above and below) that you suggest subspecific treatment in these cases.

Author Response

Reviewer #2:

This is an excellent manuscript addressing species delimitation in a polyploid complex. It follows an integrative taxonomic approach, exploring genomic, morphological, ecological and geographic  differentiation patterns, which are assessed here by several kinds of data analyses and tests, including state-of-the-art computations.

I have a few minor remarks that may aid in comprehensibility of the paper.

1)      The paper is extraordinarily long. I understand that is it methodologically complex, but the authors could consider some reduction of the Introduction and Discussion, to focus them on the most significant aspects and outcomes.

No changes here; admittedly, the paper is quite long. However, since the two other reviewers considered the length and complexity of the contribution appropriate, we refrained from shortening the text.

2)      I understand the format style of the references in the text adopted by the journal, which uses the numbers instead of the author names, but please try to avoid sentences such as: Finally, [27] presented….Most recently [19] combined….[22] considers…
It is so strange and disturbing reading such sentences.

Done; we have added author names wherever citations were part of sentences.

3)      section 2.2: please include a picture illustrating the leaf shape variation or leaf shape analysis

Done; we have included a figure with leaf silhouettes of all eight taxa (Figure 5).

4)      Table 1: It is too large and disrupts the continuity of the text. It would be more appropriate to place it as an appendix at the end of the paper.

Done; we have transferred Table 1 to the Supplementary Material (File ES04).

5)      section 3.4: it would be nice and illustrative to include a graph or table here.

Done; we have referred to Supplementary Material ES07 for plots illustrating our results.

6)      Fig. 3: please specify that the analysis is based on SNP variation. Include also taxa names not just the accession labels in the (A) graph.

Done; we have both detailed the figure caption and added taxon names to the figure.

7)      Discussion, line 517:  Figure 4 should be referred to here instead of Figure 3

Done.

8)      Discussion, lines 511-529. This appears somewhat misplaced here, in the section ‘genealogical and genetic patterns’. Maybe it could be moved or incorporated in the section ‘4.4. Integration of sources of evidence’

Done; we have transferred this paragraph to section 4.4.

9)      Discussion, lines 596-597. I completely understand and agree that the analyses aimed at finding morphological discontinuities, but I doubt these (especially regarding the leaf shape) can be considered as proxies for potential reproductive isolation.

Done; we have somewhat mitigated our statement by speaking of morphological discontinuities as potential proxies for evolutionary independence.

10)   Figure 4: please explain abbreviations in the figure legend (Eco D, Eco I, ML, MP, MN)

Done.

11)   Discussion, lines 734-742. Please, specify here (like in the paragraphs above and below) that you suggest subspecific treatment in these cases.

Done.

Reviewer 3 Report

This is the result of the experience accumulated by the research group in this plant genus. The justification and starting hypothesis are clear and well presented. A complex problem is covered by applying an integrative vision that requires a deep knowledge of the group and the application of multiple innovative techniques. The analyzes are adequate, well explained and executed, and the results obtained are precisely and clearly presented. The discussion is complete and relevant taxonomic novelties needed for this genus are concluded. Although results are may not be so relevant for a big audience, the vision and techniques applied may be of interest for any other groups of complex organisms. Overall, it was very well written and I recommend it for publication after a minor revision:

Line 96: add "and" in the sentence "...AFLP fingerprinting, and single-copy nuclear markers...".

Line 234: "an accession" instead of "a accession".

Table 1 is too large and it might be as supplementary file.

In Table 1, the locality for M60-01 and M60-011 should say "between Cuenca and Teruel" or alternatively "Cuenca-Teruel" instead of "Cuenca, Teruel" because both, Cuenca and Teruel are different provinces and it is confusing, and the geographical coordinates is a point in Cuenca province.

Line 270: Do you mean "WKM" instead of "CKM"?

In "Results" and along the ms. I suggest to say "taxon pairs" instead of "taxon combinations" when comparing no more than two taxons (I guess this occurs in all cases).

Line 323: It's better to say "...alpha level of p-value <0.01 for..." than "...alpha level of 0.01 for...".

The paragraph in lines 344-347 needs a reference to what are this results of...I guess they are results of the geographical range overlap. It also needs a reference to Table 4 where these results are presented.

I miss in the Results section a brief comment on results of on dataset of the diploid "L. vulgare-group" plus all tetraploid samples (point "b", line 258). Was this analysis  not informative at all?, Were there any relevant differences with the analysis of only all tetraploid samples?...if it wasn't relevant it would be better to eliminate from lines 258-259.

I would mark on red also sample 437-01 in Figure 2A. It would be nice to have the six groups recognized in Figure 3 also colored in Figure 2, so it is easier to follow and identify them for readers.

Last sentence of Figure 3 legend (lines 405-406) should be eliminated. This is the interpretation of the figure that is explained already in the text.

Lines 498-502: Does this means that L. virgatum might be act as a ovule donor for multiple allopolyploids?

Line 517: I guess you mean "Figure 4" instead of "Figure 3". Please, check it.

In legend of Figure 4, it would be nice to indicate that "white" means not applicable or no data...I would eliminate the border of Figure 4, so it would be a triangle.

Author Response

Reviewer #3:

This is the result of the experience accumulated by the research group in this plant genus. The justification and starting hypothesis are clear and well presented. A complex problem is covered by applying an integrative vision that requires a deep knowledge of the group and the application of multiple innovative techniques. The analyzes are adequate, well explained and executed, and the results obtained are precisely and clearly presented. The discussion is complete and relevant taxonomic novelties needed for this genus are concluded. Although results are may not be so relevant for a big audience, the vision and techniques applied may be of interest for any other groups of complex organisms. Overall, it was very well written and I recommend it for publication after a minor revision:

Line 96: add "and" in the sentence "...AFLP fingerprinting, and single-copy nuclear markers...".

Done.

Line 234: "an accession" instead of "a accession".

Done.

Table 1 is too large and it might be as supplementary file.

Done; we have transferred Table 1 to the Supplementary Material (File ES04).

In Table 1, the locality for M60-01 and M60-011 should say "between Cuenca and Teruel" or alternatively "Cuenca-Teruel" instead of "Cuenca, Teruel" because both, Cuenca and Teruel are different provinces and it is confusing, and the geographical coordinates is a point in Cuenca province.

Done.

Line 270: Do you mean "WKM" instead of "CKM"?

Done.

In "Results" and along the ms. I suggest to say "taxon pairs" instead of "taxon combinations" when comparing no more than two taxons (I guess this occurs in all cases).

Done.

Line 323: It's better to say "...alpha level of p-value <0.01 for..." than "...alpha level of 0.01 for...".

Done.

The paragraph in lines 344-347 needs a reference to what are this results of...I guess they are results of the geographical range overlap. It also needs a reference to Table 4 where these results are presented.

Done; the reference to Table 4 is now stated.

I miss in the Results section a brief comment on results of on dataset of the diploid "L. vulgare-group" plus all tetraploid samples (point "b", line 258). Was this analysis  not informative at all?, Were there any relevant differences with the analysis of only all tetraploid samples?...if it wasn't relevant it would be better to eliminate from lines 258-259.

Done; we have erased point (b) because we do not show results of the joint analysis of the L. vulgare-group and the tetraploids. This was done in a former version of the ms but turned out to be not informative.

I would mark on red also sample 437-01 in Figure 2A. It would be nice to have the six groups recognized in Figure 3 also colored in Figure 2, so it is easier to follow and identify them for readers.

Done; we have colored taxa and taxon groups found in Figure 3 also in Figure 2.

Last sentence of Figure 3 legend (lines 405-406) should be eliminated. This is the interpretation of the figure that is explained already in the text.

Done.

Lines 498-502: Does this means that L. virgatum might be act as a ovule donor for multiple allopolyploids?

No change here; this is explained and clarified in the subsequent part of the paragraph.

Line 517: I guess you mean "Figure 4" instead of "Figure 3". Please, check it.

Done.

In legend of Figure 4, it would be nice to indicate that "white" means not applicable or no data...I would eliminate the border of Figure 4, so it would be a triangle.

Done; the meaning of white cells is explained now.